# REALSEUDO FOR REAL-TIME CALCIUM IMAGING ANALYSIS

## ABSTRACT

Closed-loop neuroscience experimentation, where recorded neural activity is used to modify the experiment on-the-fly, is critical for deducing causal connections and optimizing experimental time. A critical step in creating a closed-loop experiment is real-time inference of neural activity from streaming recordings. One challenging modality for real-time processing is multi-photon calcium imaging (CI). CI enables the recording of activity in large populations of neurons however, often requires batch processing of the video data to extract single-neuron activity from the fluorescence videos. We use the recently proposed robust time-trace estimator—Sparse Emulation of Unused Dictionary Objects (SEUDO) algorithm—as a basis for a new on-line processing algorithm that simultaneously identifies neurons in the fluorescence video and infers their time traces in a way that is robust to as-yet unidentified neurons. To achieve real-time SEUDO (realSEUDO), we optimize the core estimator via both algorithmic improvements and an fast C-based implementation, and create a new cell finding loop to enable realSEUDO to also identify new cells. We demonstrate comparable performance to offline algorithms (e.g., CNMF), and improved performance over the current on-line approach (OnACID) at speeds of 120 Hz on average.

## 1 INTRODUCTION

Closed loop experiments enable neuroscientists to modify or adapt presented stimuli or introduce perturbations, such as optogenetic stimulation, in real-time based on new observations of the neural activity. Such experiments are critical for both optimizing experimental time, e.g., by optimally selecting stimuli to fit neural response models (Charles et al., 2018), or by deducing causality instead of only functional connectivity. Despite this critical need, closed loop experiments are incredibly difficult as they require real-time processing of neural data, which can be computationally intensive to process. In particular, population-level recordings using modern technologies often require significant computation to even extract individual neuronal activity traces, e.g., in high-density recordings or in fluorescence microscopy.

One particularly challenging recording technology is fluorescence microscopy, in particular multi-photon calcium imaging (CI). CI has progressed significantly since its inception with optical advances enabling larger fields of view, and therefore higher data throughput. While neuroscientists now have access to hundreds-to-thousands of neurons at a time (up to even millions), the neuronal time traces are only indirectly observed as fluorescing objects in the full image. To extract each neuron's activity, a number of methods have been developed, including matrix factorization approaches, deep learning approaches, and others (we refer to a recent review for a more complete coverage of all the available methods and their nuances (Benisty et al., 2022)).

Almost all current calcium image processing methods rely on batch processing: i.e., using a full video all at once to simultaneously identify the neurons in the data and their time traces. For example, a common approach is to identify cells in a mean image (the image containing the average fluorescence per pixel over all time) and then to extract the time-trace from the video given the neuron's location. Real-time processing does not afford such luxury. Instead, frames must be processed as they are collected. Furthermore minimal data can be stored and used, as large image batches reduce algorithmic speed. Finally, the incomplete knowledge of the full set of cells in the

video can cause unintended cross-talk as new cells firing may overlap with known cells, causing a well-documented effect of false transients (Gauthier et al., 2022).

We thus present an algorithm capable of demixing CI data frame-by-frame in real-time. Our design goals are to operate at $> 30$ Hz with minimal temporary data storage. Our primary contributions are: 1) A fast implementation of the SEUDO algorithm, 2) A new feedback loop to identify cells in real time using the SEUDO output, and 3) patch-based parallelization that enables high-throughput calcium trace estimation across larger fields of view.

## 2 BACKGROUND

Traditionally, CI analysis has been performed on full imaging videos. The goal of these algorithms is to extract from a pixel-by-time data matrix $\boldsymbol{Y} \in \mathbb{R}^{M \times T}$, where $M$ is the number of pixels in each frame and $T$ is the number of frames, a set of neural profiles $\boldsymbol{X} \in \mathbb{R}^{M \times N}$ (one for each of $N$ neurons) and a corresponding set of time traces $\boldsymbol{\Phi} \in \mathbb{R}^{N \times T}$. The former of these has, as each column of $\boldsymbol{X}$, a single component profile depicting which pixels constitute that fluorescing object, and how strong that pixel is fluorescing. The latter has as each row the corresponding time traces that represent how bright that object was at each frame. These time-traces are particularly important for relating neural activity to each other via population dynamics, or to stimuli and behavior.

In typical approaches, full videos are required to either 1) identify summary images (e.g., mean or max images (Pachitariu et al., 2013; Diego & Hamprecht, 2014; Petersen et al., 2018; Klibisz et al., 2017)) to identify cells in, 2) to create a dataset within which points are clustered into cells (Kaifosh et al., 2014; Spaen et al., 2019; Mishne et al., 2018; Reynolds et al., 2017; Apthorpe et al., 2016; Soltanian-Zadeh et al., 2019; Bao et al., 2021; Soltanian-Zadeh et al., 2019; Kirschbaum et al., 2020), or 3) to perform simultaneous cell identification and demixing (Pnevmatikakis et al., 2016; Pachitariu et al., 2016; Charles et al., 2022; Haeffele & Vidal, 2019; Inan et al., 2017; Maruyama et al., 2014; Mishne & Charles, 2019; Song et al., 2017; Giovannucci et al., 2019) (e.g., via matrix factorization or dictionary learning). For example, in the latter example, the data decomposition is solved via a regularized optimization, e.g.,

$$\widehat{\boldsymbol{X}}, \widehat{\boldsymbol{\Phi}} = \arg \min_{\boldsymbol{X}, \boldsymbol{\Phi}} \|\boldsymbol{Y} - \boldsymbol{X}\boldsymbol{\Phi}\|_F^2 + \mathcal{R}_X(\boldsymbol{X}) + \mathcal{R}_\Phi(\boldsymbol{\Phi}), \tag{1}$$

where $\|\cdot\|_F^2$ is the Frobenius norm (sum of squares of all matrix elements), and $\mathcal{R}_X(\boldsymbol{X})$ and $\mathcal{R}_\Phi(\boldsymbol{\Phi})$ are regularization terms for the profiles and time-traces, respectively. While many regularization combinations exist, common terms include sparse neural firing, sparse overlaps, non-negativity, and spatial locality. Regardless, all methods require all frames to identify the fluorescing components, with the exception of OnACID (Giovannucci et al., 2017) and FIOLA Cai et al. (2023).

OnACID and FIOLA operate in an on-line manner, utilizing the buffer of last $l_b$ residuals $\boldsymbol{r}_t = \boldsymbol{y}_t - \boldsymbol{X}\boldsymbol{c}_t - \boldsymbol{B}\boldsymbol{f}_t$ where $\boldsymbol{X}$ and $\boldsymbol{c}$ represent the spatial and temporal profiles of already recognized cells and $\boldsymbol{B}$ and $\boldsymbol{f}$ represent the spatial and temporal profiles of the known background signal. Both methods use a local Constrained Non-negative Matrix Factorization (CNMF) (Pnevmatikakis et al., 2016) in the spatial and temporal vicinity of that point. CNMF is an off-line algorithm that repeatedly performs alternating optimizations on $[\boldsymbol{X}, \boldsymbol{B}]$ and on $[\boldsymbol{c}, \boldsymbol{f}]$ using the full dataset, until it converges to a designated precision. Both methods require initialization periods, and FIOLA further requires GPU and CPU optimizaiton, raising the computational infrastructure costs. We seek a solution that does not need any initialization data and can be run on simpler CPU machines for easier incporporation into user's workflows.

### 2.1 SPARSE EMULATION OF UNKNOWN DICTIONARY OBJECTS

One primary challenge in fully on-line settings is the incomplete knowledge of all fluorescing components at the experiment onset. Even in off-line methods, incomplete identification of cells can create scientifically impactful cross talk—termed *false transients*—in inferred activity (Gauthier et al., 2022; Inan et al., 2017). Another challenge is identifying new components from few frames: ideally from individual frames to reduce memory usage. Recent work has provided an algorithm that solves both challenges: The Sparse Emulation of Unused Dictionary Objects (SEUDO) algorithm (Gauthier et al., 2022).

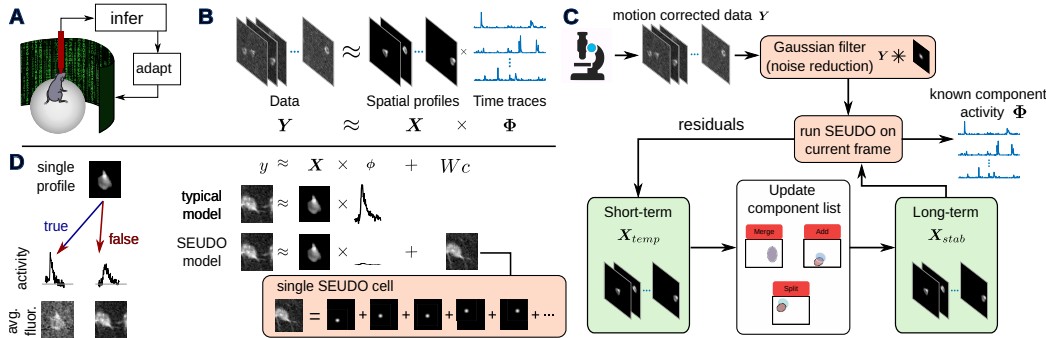

Figure 1: The realSEUDO algorithm. A: Real-time inference of cells and their activity from calcium imaging is crucial to closed-loop experiments. B: Typical calcium imaging demixing requires batch processing. C: We propose a method based around the SEUDO estimation algorithm that can identify cells in real time by robustly removing known cells and using the residuals to identify new cells in the data. D: realSEUDO builds on the SEUDO algorithm that is robust to missing cells. SEUDO was developed to prevent false activity in known cells from unknown cells (left) by explicly modeling contamination as a sparse sum of small Gaussian blobs (right). SEUDO cells further provide an approximation of the missing, structured fluorescence in the frame.

SEUDO is a robust time-trace estimator for neuronal time traces. Given a single fluorescence video frame $y_t$, and a set of known profiles $X$, SEUDO models contamination as $Wc$ where $W$ is a basis of small Gaussian bumps that linearly construct the interfering components, weighted by the sparse coefficients $c$ (i.e., most $c$ values are zero). SEUDO then solves the optimization

$$\widehat{\phi}_t = \arg \min_{\phi, c \geq 0} \left[ \min \left[ \|y_t - X\phi_t\|_2^2, \|y_t - X\phi_t - Wc\|_2^2 + \lambda\|c\|_1 + \gamma \right] \right], \qquad (2)$$

where $\lambda$ and $\gamma$ are model parameters and the internal min selects from the two internal expression that which has the minimal value. Since SEUDO operates per-frame, $y_t$, $Phi$, $c$ are all vectors. While SEUDO has demonstrated the ability to remove false transients (Gauthier et al., 2022), SEUDO's application has been limited to off-line post-processing due to: (1) slow computational speed, and (2) the need for pre-defined profiles $X$.

## 2.2 THE FISTA ALGORITHM

The computational bottleneck in SEUDO is a weighted LASSO (Tibshirani, 1996) optimization, which can be implemented with the Fast Iterative Shrinkage-Thresholding Algorithm (FISTA) that implements a momentum gradient descent (Beck & Teboulle, 2009). FISTA optimizes

$$\min \left[ F(x) \equiv f(x) + g(x) : x \in \mathbb{R}^n \right], \qquad (3)$$

where $f(x)$ is a smooth convex function with Lipschitz constant $L > 0$, such that $\|\nabla f(x) - \nabla f(y)\| \leq L\|x - y\| \forall x, y \in \mathbb{R}^n$, e.g. in SEUDO $f(x) = \|y_t - X\phi_t - Wc\|_2^2$, and $g(x)$ is a continuous convex function that is typically non-smooth, e.g., $g(x) = \lambda\|x\|_1$. Each descent step of FISTA consists of an ISTA descent step and a momentum step:

$$x_k = \arg\min_x \left[ g(x) + \frac{L}{2} \left\| x - \left( y_k - \frac{1}{L}\nabla f(y_k) \right) \right\|^2 \right] \qquad (4)$$

$$y_{k+1} = x_k + \eta_k(x_k - x_{k-1}), \qquad (5)$$

where the parameter $\eta_k$ gradually reduces with $\eta_k = \frac{t_k - 1}{t_{k+1}}$, $t_{k+1} = \frac{1 + \sqrt{1 + 4t_k^2}}{2}$, $t_1 = 1$.

## 3 REAL-TIME SEUDO

Here we develop Real-time SEUDO (realSEUDO) that resolves the primary limitations of SEUDO and extends the algorithm significantly from a time-trace estimator to a real-time cell identification

method. Specifically we improve the computationally intensive momentum descent algorithm used to solve Equation equation 2 by reducing the number of steps of momentum descent, implementing parallelism, optimizing internal computations (e.g., of smoothness parameters), and reducing the complexity of the original fitting problem without a substantial loss of quality by manipulating it inputs. Moreover we add a new algorithm that automatically recognizes the neurons that have not previously been seen and adds them to the SEUDO dictionary. Finally, we present a patch-based parallelism that avoids the slowdown of running LASSO in higher dimensions.

realSEUDO 1 operates at high level as follows (Fig. 1): For simplicity we assume that we begin with no known profiles. When significant activity occurs, triggering a SEUDO cell, that cell is stored in a temporary array. Profiles are moved from the temporary profiles to the static set of profiles when they disappear from the current frames. The static profile set is then used to identify the activity of those components in future frames, with unexplained components becoming SEUDO cells and cycling back into the temporary profiles, followed by an update of the static profile set.

## 3.1 SEUDO OPTIMIZATION

Our first step in creating realSEUDO is to vastly increase the speed of SEUDO to operate at $> 30$ fps. The original implementation of SEUDO relied on the TFOCS library (Becker et al., 2011) in MATLAB. We began by improving the run-time by switching from MATLAB's interpreted programming language running TFOCS to a fast implementation of LASSO (Tibshirani, 1996) via FISTA (Beck & Teboulle, 2009) in the C++ compiled language. To further improve performance, we optimized the C++ code with the use of templates to eliminate the function call overhead in tight loops, and also employ parallelism, based on the POSIX threads with TPOPP library wrapper (Babkin, 2010). To prevent a bottleneck from the passing of data through Matlab's OOP API at the MATLAB/C++ interface, we switched to the non-OOP version of the MATLAB-to-C API.

While beneficial, simply coding SEUDO in C++ was not sufficient to achieve the desired processsing rate. We further improved runtimes with optimizations to the computation of the cost function and derivatives. The partial LASSO component of SEUDO that performs optimization on a single frame $y$ using the FISTA algorithm can be written as $\arg\min f(\psi) + \lambda g(\psi)$, where $f = \|y - \chi\psi\|_2^2$ is the least-squares portion and $g = \|\psi\|_1$ is the $\ell_1$ penalty. In the FISTA algorithm the spaces of time traces and Gaussian kernels are unified, with $\chi$ representing a concatenation of $X$ and $W$, $\psi$ representing a concatenation of $\Phi$ and $c$. With the number of pixels in the frame $M$, number of neurons $N$, and number of Gaussian kernels $K$, $y \in \mathbb{R}^M$, $\chi \in \mathbb{R}^{M \times (N+K)}$, $\psi \in \mathbb{R}^{N+K}$, $\lambda \in \mathbb{R}^{N+K}$, with the first $N$ elements of $\lambda$ corresponding to $\Phi$ being equal to 0.

In working with this optimization program, a number of internal computations become bottlenecks: specifically computing the gradients $\nabla_\psi f$ and $\nabla_\psi(f + g)$, the Lipshitz smoothness estimation, and the momentum/stopping criteria.

We reduce the burden of the gradient computations by both reducing the number of times the gradient must be used, and by improving the internal gradient computation. For the former, we note that naive implementations compute both a step in the direction of $\nabla_\psi f$ and then in the direction of $\nabla_\psi(f + \lambda g)$. Moreover, we noted that these two steps in slightly different directions cause the gradient to dither around the optimum. We thus instead only take a step in the direction of $\nabla_\psi(f + \lambda g)$ (similar to (Beck & Teboulle, 2009)). For the latter, computing $\nabla_\psi(f + \lambda g)$ requires matrix vector multiplications with $\chi$ and its transpose. Since $\chi$ is sparse, to save memory and computation, we avoided storing the $\chi$ matrix explicitly, instead generating it on the fly in an efficient convolution. Specifically, the gradient $\nabla_\psi f$ requires computing $\chi^T \chi \psi$, which we reorganize to compute in two passes: The first pass:

$$v_j = \sum_{1 \le i \le N+K} y_j - \chi_{ji}\psi_i, \qquad \frac{\mathrm{d}f}{\mathrm{d}\psi_m} = 2 \sum_{1 \le j \le M} \chi_{jm}v_j, \qquad (6)$$

the first computes intermediary variables, and the second uses these values to compute the gradient dimensions.

This split into two passes factors out repeated computations, and reduces the time complexity from $O(n^3)$ to $O(n^2)$. Moreover, the computation of each pass becomes highly parallelizable, with partitioning of the first pass by $j$ and of the second pass by $m$, and efficiently skipping the iterations over the zero elements in the sparse matrix $\chi$.

To improve the efficiency of estimating the Lipschitz constant $L$, note that $L = \max(\|\nabla f(\boldsymbol{x}_1) - \nabla f(\boldsymbol{x}_2)\|/\|\boldsymbol{x}_1 - \boldsymbol{x}_2\|)$. We found that for our cost function, we could approximate $L$ with independent computations in each dimension. This estimation reduced the number of steps by as much as 30% over typical computations of $L$ before each step based on the local gradient.

The momentum descent central to solving the partial LASSO tends to spend many steps on stopping the momentum, especially with the large values of the Lipshitz constant $L$ (i.e. the non-momentum steps are small). One such case is "circling the drain" around the minimum, with the momentum causing the overshoot in one dimension while another dimension is stopping. Another case is when a dimension is moved past the boundary (e.g., $\boldsymbol{x} \geq 0$ for SEUDO). The momentum continues to push the solution to negative values, producing suboptimal solutions and increasing the number of steps necessary. FISTA includes a parameter $\eta$ that progressively limits the top speed of descent to reduce such problems. We improved these cases by resetting the momentum to zero on a dimension when it either attempts to cross into the negative values or when its gradient changes sign. The dimensional momentum stopping stops abruptly at the right time, obviating the need for slowing and thus we can simplify FISTA by fixing $\eta = 1$.

We have also experimented with using the dimensional momentum stopping on a different kind of optimization problems, on momentum optimization of the neural network training, where it provided a substantial improvement. While FISTA performed worse than simple gradient descent, modified FISTA produced the same error rate and squared mean error in about 10 times fewer training passes than simple descent, or about 10 times lower error rate and 1.5 times lower mean square error in the same number of passes. The FISTA algorithm incorporating all these extensions is presented in Algorithm 2 in Supplement.

As a third step, we modified the optimization program (the SEUDO model) itself to achieve the final speedups. In the original SEUDO, Gaussian components in $\boldsymbol{W}$ are spaced apart by one pixel. We instead found that this tight spacing is redundant. The kernels with radius $r$ cover $(\pi * r^2)$ pixels, and thus each pixel is covered by $(\pi * r^2)$ kernels with varying weights. This redundancy results in FISTA continuing to adjust the kernel coefficients $\boldsymbol{c}$ after the neural activations $\boldsymbol{x}$ converge. Reducing the number of kernels thus reduces both the number of gradient descent steps and the per-step cost, accelerating the computation more than quadratically. For a kernel with diameter of 30 pixels this improves the performance by a factor of over 100 without substantial degradation of false transients removal or the recognition of the interfering components' shape $\boldsymbol{W}\boldsymbol{c}$.

We tested our speedups against the original SEUDO on 45000 frames across 50 cells from Gauthier et al. (2022). SEUDO ran at 5.8-6.9 s/cell on a Macintosh M1. The optimized C++ implementation without the MATLAB C++ API reduced the runtime to 0.9-1.1 s/cell (a 6-7x improvement). Sparse SEUDO provided further acceleration to a run time of 0.2 s, with 0.1 s for the computation and 0.1 s for the overhead of converting data between Matlab and native code; a total of a 29-34.5x speed-up.

## 3.2 Automatic cell recognition

With our improvements, the core SEUDO algorithm was able to run in real time, and we next moved on to the cell recognition feedback loop that forms the realSEUDO algorithm. As realSEUDO must rely only on past video frames to identify new profiles, we designed realSEUDO to run on a frame-by-frame basis. At a high level, our automatic cell recognition first runs SEUDO on the current incoming (denoised) frame given the known profiles $\boldsymbol{X}_{stab}$ identified thus far. The loop then identifies contiguous bright areas in the residual frames, i.e., the SEUDO cells resulting from SEUDO, and places them in a 'temporary profile' array $\boldsymbol{X}_{temp}$. These temporary profiles are then updated (via merging with new potential profiles) given new, incoming, frames until they are stable and moved to the stable, known profile list $\boldsymbol{X}_{stab}$ that is updated less frequently by addition, merging and splitting of temporary profiles.

Procedually, we first preprocess each incoming frame with the goal of reducing noise to improve profile recognition. Typical calcium imaging analysis often uses running averages in space and time for noise reduction. We thus implement both a spatial Gaussian filter, as well as a running average of several sequential frames. We keep the window length as a tunable parameter that can be set to one to perform frame-by-frame processing and minimize temporal blurring of neural activity.

After each frame is denoised, we estimate the activation level for each of the stable profiles $\boldsymbol{X}_{stab}$ using the fast version of the SEUDO estimator. We denote the activation level for the $k^{th}$ profile at time $t$ as $\phi_{kt}$. SEUDO also returns a robust residual that contains all the fluorescence that was not captured by $\boldsymbol{X}_{stab}$. We then run the residual through SEUDO a second time using the temporary profiles $\boldsymbol{X}_{temp}$ to test if any temporary profile represents the fluorescence well and should be moved to $\boldsymbol{X}_{stab}$ from $\boldsymbol{X}_{temp}$. The residual after this second application of SEUDO represents completely unknown profiles and are analyzed separately to determine if a new member of $\boldsymbol{X}_{temp}$ should be created.

Starting from the bottom up, the detection of new temporary profiles is based on finding the areas of the image that stand above the noise level. The noise level is evaluated by noting that most of each video frame has no activity, indicating that the median pixel value will be very close to the median value of the pixels in an all-dark frame containing the same noise. The half-amplitude of the noise $\sigma_{ha}$ can be estimated as:

$$\sigma_{ha} = \text{median}(\boldsymbol{y}_t) - \min(\boldsymbol{y}_t). \tag{7}$$

In some cases different areas of the image may contain a different amount of background lighting (e.g., changes in neuropil), which can skew the noise estimate. To overcome this challenge, we split the larger image into sections, with each section computing a local median which is smoothly interpolated between section of the image. This can be thought as either a krigging procedure or a cheap approximation to a local median evaluated independently for each pixel in the image.

To merge new profiles into the exiting profiles when adding new profiles to $\boldsymbol{X}_{temp}$, we compute an overlap score. The overlap computation is not a plain spatial overlap but includes a heuristics that identifies when one profile is mostly contained in another. The condition for merging two profiles in $\boldsymbol{X}_{temp}$ is based on the comparison of numbers of common and unique pixels between profiles, where $P_1$ and $P_2$ are numbers of pixels in each of two profiles, $U_1$ and $U_2$ are the numbers of unique pixels in each profile, $C$ is the number of common pixels, $B_1$ and $B_2$ are the perimeters of bounding boxes for each profile, and $k_{temp}$ is a constant with an empirically chosen value of 0.75:

$$U_1 \leq B_1 * 0.5 \quad \text{or} \quad U_2 \leq B_2 * 0.5 \quad \text{or} \quad C \geq k_{temp} * \min(P_1, P_2) \tag{8}$$

After a profile is moved from $\boldsymbol{X}_{temp}$ to $\boldsymbol{X}_{stab}$, a different score is computed pair-wise between the new profile and each existing profile, to decide whether they should be left separate, or merged, or one of profiles split. If a merge or split is performed, the original profiles are removed from $\boldsymbol{X}_{stab}$ and the results entered recursively into $\boldsymbol{X}_{stab}$ as new profiles. The score for two profiles $A$ and $B$ in $\boldsymbol{X}_{stab}$ is computed based on the brightness and measures of least-squares fit of the cells into each other 1) as whole cells (i.e., $\alpha_{AB} = \langle A, B \rangle / \langle A, A \rangle$) and 2) using only the overlapping region (i.e., $\beta_{AB} = \langle A_{ol}, B \rangle / \langle A_{ol}, A_{ol} \rangle$ where $A_{ol}$ is the profile $A$ restricted to the region overlapping with $B$). $\beta_{AB}$ is used as a measure of difference in brightness, against which the fit of the whole cells $\alpha_{AB}$ is compared as a measure of proximity in shape. Specifically we compute two ratios $\rho_{AB}$ and $\rho_{BA}$ are computed as

$$\rho_{AB} = \frac{\alpha_{AB}}{\beta_{AB}}, \qquad \rho_{BA} = \frac{\alpha_{BA}}{\beta_{BA}}. \tag{9}$$

A higher value of $\rho_{AB}$ (always $\leq 1$) means that cell A fits better inside cell B. The value of 1 means that it fits entirely inside cell B. The same principle applies symmetrically to $\rho_{BA}$.

The scores express the following logic: If two temporal profiles are a close match in cross-section, they likely represent the same cell and should be merged. If they overlap only partially, they likely represent separate cells. If one cross-section is inside the other, look at relative brightness: if the smaller cross-section is also weaker, it's likely a weaker partial activation of the same cell and should be merged, if the smaller cross-section has a close or higher brightness, the larger cross-section likely represents an intertwining of two cells that has to be split.

### 3.3 PATCHING AND PROFILE MATCHING

realSEUDO, although highly efficient for smaller patches, is still based on the LASSO algorithm that reduces in efficiency with much larger frames. Thus we adopt a patching scheme that breaks each frame into small patches that can be parallelized to maintain the high framerate by utilizing the multi-threading in many modern processors. Patching, however, requires matching profiles across patches. Traditionally profiles discovered in data split into patches is to add overlap margins to the patches

and to use profiles overlap in this region to determine matchings in neighboring patches. Additional margins, however, introduces redundant computation and decreases computational efficiency.

In realSEUDO we note that the logic behind the scoring we use to merge profiles in $X_{stab}$ and $X_{temp}$ within each patch can also be used to score the match of profiles across patches. Specifically, we extend the matching to include the profile temporal activity as an additional dimension to find matching cells in neighboring patches via consecutive gluing of the profiles. The highest score is assigned to the bidirectional match of both spatial and temporal dimensions, a lower score to symmetrical match of spatial dimensions and asymmetrical match of temporal dimension, a yet lower score to an asymmetrical match of both kinds of dimensions. We have observed successful matches even with zero margin, using the neighboring strips of pixels around the perimeters of the patches as the spatial dimension for matching.

## 4 RESULTS

**Simulated data experiments:** We first applied all three algorithms to a simulated video created with Neural Anatomy and Optical Microscopy simulation (NAOMi) (Song et al., 2021). Specifically we simulated the neural activity over 20000 frames at 30Hz with fame size of 500x500 pixels. There are approximately 450 cells visible in this dataset (i.e. fluorescing cells intersecting the plane of imaging). We benchmarked the patch-based parallel processing of realSEUDO 80x80 pixel patches. For comparison we ran the off-line CNMF (a staple batch-based calcium imaging demixing algorithm) and OnACID, the computationally similar on-line method. realSEUDO found 201 true cells, identified as strongly correlated with ground-truth cells, while OnACID found 152 and CNMF (the offline method) found 308 (Fig. 2A-B). Furthermore, both realSEUDO and OnACID found many fewer false positives than CNMF, presumably because they cannot be fooled by small fluctuations integrated over the full recording (Fig. 2C-D). Note that to remove the confound of post-processing we followed prior work Song et al. (2021) in using the CNMF raw fluorescence traces instead of the model-based denoised traces. On average, realSEUDO processed 67.8 frames per second end-to-end, while OnACID ran at 9.2 fps.

**Applications to _in-vivo_ mouse CA1 recordings** We applied realSEUDO to an _in vivo_ calcium imaging recordings from mouse hippocampal area CA1 previously described in Gauthier et al. 2022 (Gauthier et al., 2022). They consisted of 36 videos, each sized 90x90 pixels with 41750 frames sampled at 30 Hz. The outputs had previously been verified manually by Gauthier et al. 2022 (Gauthier et al., 2022) with human labeling of CNMF outputs. We applied realSEUDO to all videos and compared the outputs with the current online cell demixing algorithm OnACID (Giovannucci et al., 2017), as well as a popular offline algorithm, CNMF (Giovannucci et al., 2019), as an additional baseline.

We benchmarked realSEUDO against OnACID (an online analysis tool) and CNMF (an offline analysis tool) on real in-vivo calcium imaging movies (Fig. 3). Algorithmic performance was measured on an x86-64 computer with 48 CPU cores (Intel Xeon 6248R), 2 hyperthreads per core, 78 GB of memory, and without the use of a GPU. The initialization times were not included. On average, realSEUDO processed 162 frames per second compared to 26 processed by OnACID and 13 by CNMF: an improvement of 6.5x and 12.5x respectively (Fig. 3C). Quality-wise, we found that OnACID exhibited difficulties with adapting to larger ranges of pixel brightness, sometimes missing bright cells. Scaling pixel values improved OnACID results, but only mildly (one additional cell). Numerically OnACID and CNMF appear similar but they identified different components. SEUDO results were most similar to CNMF, and with additional cells identified, and less false positives (Fig. 3B). Finally, the per-transient manual classification provided by (Gauthier et al., 2022) enabled us to assess if realSEUDO inherited the false transient removal properties of SEUDO. For a reasonable value of $\lambda = 0.15$, realSEUDO had a true positive rate of 75% and a false positive rate of 24%. While these numbers are a bit lower than the numbers reported in (Gauthier et al., 2022), in that study the authors average N=3 frames to reduce noise, while maintained single-frame analysis. This can be evident by the fact that missed transients were very small: realSEUDO kept 98% of real fluorescence and only 15% of false fluorescence.

**Additional _in-vivo_ tests:** As final test we applied all three algorithms (realSEUDO, OnACID, CNMF) to a 2000-frame mesoscope video example collected by the Yuste lab at Columbia Uni-

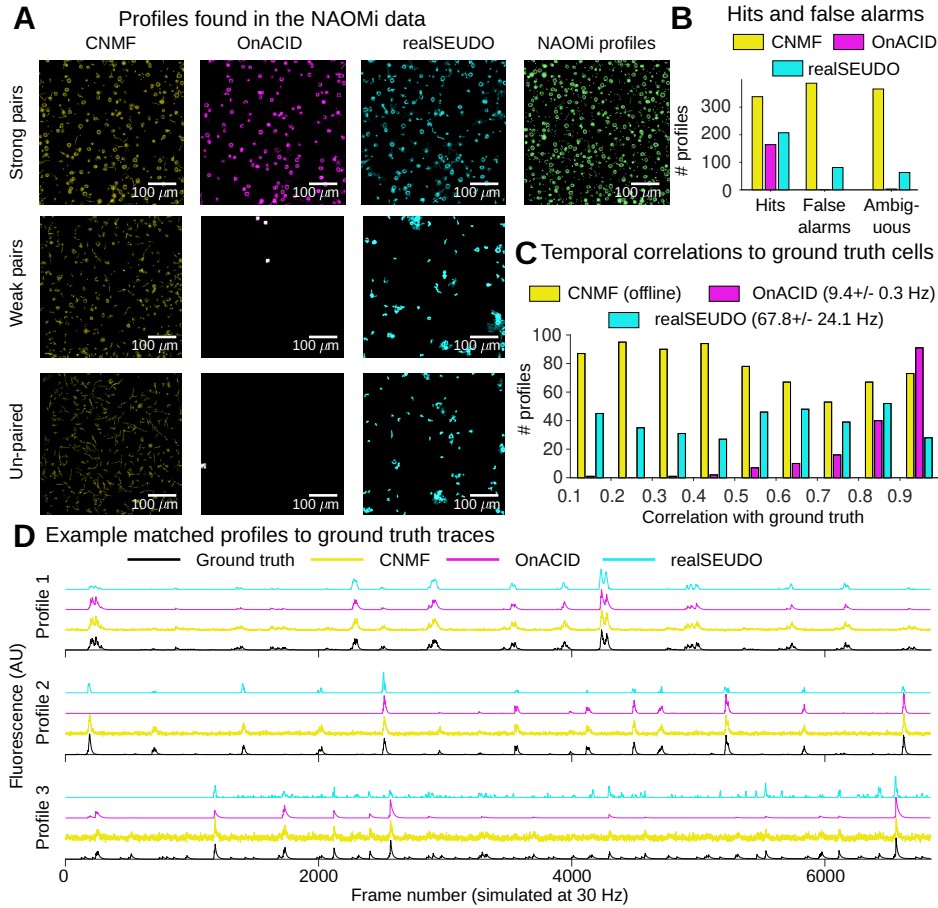

Figure 2: NAOMi results: A) Found cells in NAOMi for CNMF, OnACID and realSEUDO separated into Hits (strong or weakly correlated) and false alarms (uncorrelated). B) realSEUDO finds more cells than OnACID with minimal false positives. C) Temporal correlations for found "hits". D) Examples time-traces show correlation to ground truth.

versity and provided with the OnACID github package as a demo. For this example we similarly saw improved cell detection and runtime improvement in terms of fps (Fig. 3B).

## 5 DISCUSSION

We present here an online method for cell detection and fluorescence time-trace estimation from streaming CI data: realSEUDO. realSEUDO is based on the SEUDO robust time-trace estimator that reduces bias due to unknown cells while also providing approximate shapes of the unknown fluorescing objects. To build realSEUDO we 1) improved SEUDO's slow runtime via significant modifications at the code, algorithmic, and model levels 2) built a new feedback loop that allowed SEUDO (which has no cell finding component currently) to identify cells in real-time. Overall, realSEUDO can achieve frame processing rates of 80-200 fps, depending on cell density. While our goal was to exceed the typical 30 Hz data collection rate common to many experiments, the high processing efficiency leaves additional time to compute feedback in future closed loop systems. Moreover, realSEUDO can scale with faster recording rates as calcium indicators become faster, e.g., GCaMP8 (Zhang et al., 2023).

realSEUDO's implementation exhibits a higher degree of parallelism than OnACID, however both are likely constrained by the employed tools. Specifically, the measurements in Fig. 3C show very little fps fluctuation with respect to cell count for OnACID, to the point where the error bars are

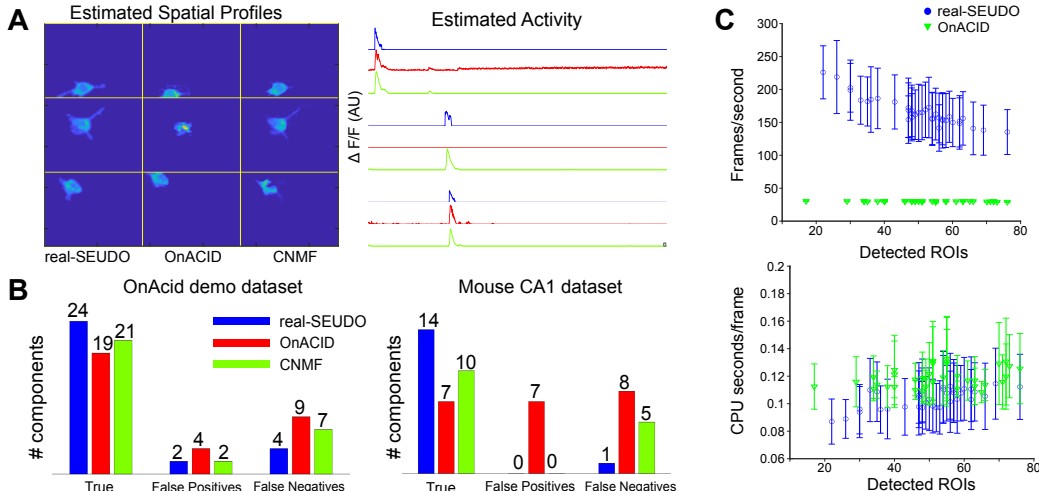

Figure 3: (A) Selected cell profiles and time-traces generated by realSEUDO, OnACID, and CNMF respectively on a subset of 2000 frames from a single image patch. (B) Counts of true positive, false positive and false negative cells found by each algorithm for two different recordings: all 41,750 frames data from one video from Gauthier et al. (Gauthier et al., 2022) (right) and from the OnACID demo (left). (C) Top: Total computational performance as a function of the number of detected cells for realSEUDO and OnACID, evaluated on the full set of 36 movies from Mouse CA1. Bottom: CPU use in CPU seconds per frame as a function of the number of detected cells for realSEUDO and OnACID, evaluated on the same 36 recordings.

invisible, which might be due to a bottleneck in a single thread. realSEUDO has a lower latency to the first events for a new cell, OnACID requires a history of 100 frames to recognize a cell, while realSEUDO would produce the first events starting with the first frame that passes the low brightness threshold. Alternatively, OnACID has a higher native scalability with respect to the frame area, conditioned on similar cell counts. This likely result from OnACID's algorithm restriction of processing to areas in the immediate vicinity of known cells. realSEUDO can still achieve a high processing efficiency with reasonable hardware with our parallelization. Future research may further improve realSEUDO runtime by blanking out entire patches until activity is detected via simpler activity detectors.

One strength of realSEUDO is that it can be initialized with either an empty profile set or a pre-computed set, e.g., via prior analysis or via anatomical (z-stack) imaging. The ability to start from nothing will be useful in mesoscope settings when the field-of-view may be changed on-the-fly, and not requiring an initialization step will reduce start-up overhead. Furthermore, we note that many of our speed adaptions deviate from the traditional gradient descent approach. In particular the approximation of the Lipshitz constant and the changes in the momentum and stopping criteria. We have observed that in other domains, in particular for training deep neural networks, these deviations appear to also provide significant speedups. We thus will consider future research directions in analyzing and quantifying the extent of these improvements across broader applications.

**Limitations:** While our results achieve the design criteria we initially set out, there are some potential barriers. For one, as with many real-time systems, the compute environment is very important to configure correctly. We have found the importance of explicitly setting the Linux CPU manager to enable the performance mode. The default automatic adaptable mode does not react properly to CPU loads of less than 100% of the whole capacity, and significantly skews the benchmarking by running the CPUs at low frequency.

While our core algorithm is written completely in C++, and thus open source, we have found MAT-LAB convenient and efficient as a wrapper for prototyping wrappers for our core functions. Further work will add Python wrappers to allow for seamless integration into both MATLAB and Python pipelines, enabling realSEUDO to be more widely used. Finally, we focused here only on cell detection, assuming access to the on-line motion correction algorithm from OnAcid. Future work should more holistically incorporate the full pipeline in C++.

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
