# OpenReview forum: "realSEUDO for real-time calcium imaging analysis"
_ICLR.cc/2024/Conference — ICLR 2024 Conference Withdrawn Submission_

### Official Review · Reviewer_vjPy · 2023-10-22

**Soundness:** 3 good
**Presentation:** 3 good
**Contribution:** 2 fair
**Rating:** 3
**Confidence:** 3

**Summary:**

This paper presents a software suite for real-time extraction of neural activity from calcium imaging video data for a neural population. The general problem is one of identifying pixel/voxels that correspond to a single neuron, and the CI intensity time profile of that neuron, over all neurons in the population that has been imaged in a CI video. The difficulty of the problem is well known since neurons very close to each ther (and therefore overlapping on pixels) tend to spike approximately synchronously, and therefore their calcium imaging intensity profiles have major temporal overlaps. The problem therefore requires both clustering of pixels (corresponding to a single neuron) and demixing of time profiles of these neurons. Various objectives have been proposed over the years with different regularization terms.

This paper starts with the objective proposed by SEUDO (Sparse emulation of unused dictionary objects) which minimizes the minimum of two terms, one an L2, and the second a sum of an L2 and L1 term. Essentially making it a weighted LASSO problem. The paper then uses the FISTA (fast iterative shrinkage thresholding algorithm) algorithm for the lasso problem to solve the chosen seudo CI objective function.
The paper then throws in many additional software tweaks. 1) reducing the number of momentum descents 2) adding parallellism 3) optimizing smoothness parameters, etc., to get it to a point where the optimization can be done in real time,

**Strengths:**

With SEUDO+FISTA+ all the additional tweaks that the authors have introduced, they do manage to solve the problem on-line from a 30 frames per second CI video. This is demonstrated both in simulated data and a real in-vivo CA1 dataset, and compared against a competing technique ONAcid.

**Weaknesses:**

Unfortunately, all the contributions of this paper are tweaks to the already present SEUDO and FISTA; the tweaks are all listed in section 3. These include, using a fast implementation of lasso using fista in a c++ library, solving the problem in a patch based manner in parallel (since pixels corresponding to a neuron are naturally localized in a video frame), speeding up a gradient calculation which requires computing the product of a matrix with its transpose by splitting things up into two steps and assuming the matrix is sparse, realizing that gaussian kernel in seudo does not have to be too small and can be enlarged without loss of accuracy, and many many more.

Each tweak can be justified only under particular situations, and moreover it was very difficult to tell what the contribution of each tweak was to the overall speed up of the technique.

**Questions:**

It would be very helpful if the authors could run a profiler where they turn each tweak on and off and run it on several simulated datasets, so that the contribution of each tweak to the overall performance becomes clear.

---

### Official Review · Reviewer_11J3 · 2023-10-30

**Soundness:** 3 good
**Presentation:** 3 good
**Contribution:** 2 fair
**Rating:** 5
**Confidence:** 4

**Summary:**

The paper presents a new method, realSEUDO, that improves upon existing methods (e.g. CNMF, OnACID, SEUDO) for online analysis of calcium images. It runs faster than previous methods and can better handle missing cells. It is primarily based on the time-trace estimator SEUDO method.

**Strengths:**

It is a great benefit to require no initialization (empty stack) to begin to run this algorithm online.

It is an evident strength to be able to run at faster framerates (>100 fps in some benchmarks).

Paper is well-written and figures demonstrating benchmarks are clear.

**Weaknesses:**

Numerous typos (e.g. 'optimizaiton', 'incporporation') should be addressed with a spellchecker.

The explanation of the method is a bit lacking. While it does draw heavily on previously published methods, more description and intuition beyond copying the formulas (Eqns 2-5) should be given.
The main text would also benefit from an overall algorithm, because it became quite confusing where each of the authors' iterative improvements fit into the method as a whole.

Only one real dataset was used for comparison, with quite small image sizes, and it did not appear to be an existing community standard (e.g. neurofinder or previously manually annotated datasets). It would be good to include these comparisons as well.

**Questions:**

Could the authors clarify what they mean by 'FIOLA further requires GPU and CPU optimizaiton [sp], raising the computational infrastructure costs'? What optimization and how does it increase costs?

For the new cell finding from residuals, could the authors provide a comparison with how e.g. OnACID handles this? Not just a comparison of the results, but of the approach.

**Details Of Ethics Concerns:**

I suspect this is just a case of one author re-using their own figures, versus copying from someone else's work, but since this is blind I figured I'd mention it anyway. In this paper Figure 1 B is a direct copy of another paper's Figure 1 A (https://ieeexplore.ieee.org/stamp/stamp.jsp?tp=&arnumber=8683375, G. Mishne and A. S. Charles, "Learning Spatially-correlated Temporal Dictionaries for Calcium Imaging," ICASSP 2019 - 2019 IEEE International Conference on Acoustics, Speech and Signal Processing (ICASSP), Brighton, UK, 2019, pp. 1065-1069, doi: 10.1109/ICASSP.2019.8683375.).
I would consider this minor but needing to be addressed by the authors.

---

### Official Review · Reviewer_26AX · 2023-11-01

**Soundness:** 3 good
**Presentation:** 3 good
**Contribution:** 3 good
**Rating:** 6
**Confidence:** 2

**Summary:**

The manuscript proposes a new real-time processing workflow for multi-photon calcium imaging (CI). The proposed workflow turns the Sparse Emulation of Unused Dictionary Objects (SEUDO) algorithm to a new on-line processing algorithm that simultaneously identifies neurons in the fluorescence video and infers their time traces. The experiments demonstrate comparable performance to offline algorithms (e.g., CNMF), and improved performance over the current on-line approach (OnACID).

**Strengths:**

1. The manuscript gives very detailed implementation information on how to achieve online processing speed.

2. The impact of real-time inference of neural activity from streaming recordings is significant in neuroscience.

3. The motivation and the existing methods are articulated very clearly.

**Weaknesses:**

1. The manuscript is mainly about documenting implementation tricks. The changes are rather heuristic, e.g. reducing the number of times the gradient is used, without too much theoretical justification. I'm sure it works in practice, but the methodological contribution is incremental in the conventional sense.

2. It is unclear why the authors would want to seek a solution that does not need any initialization data and can be run on simpler CPU machines. These are no longer bottlenecks in today's research labs. GPUs are easily accessible.

**Questions:**

N/A